# OPTIMIZED MINIMAL 4D GAUSSIAN SPLATTING

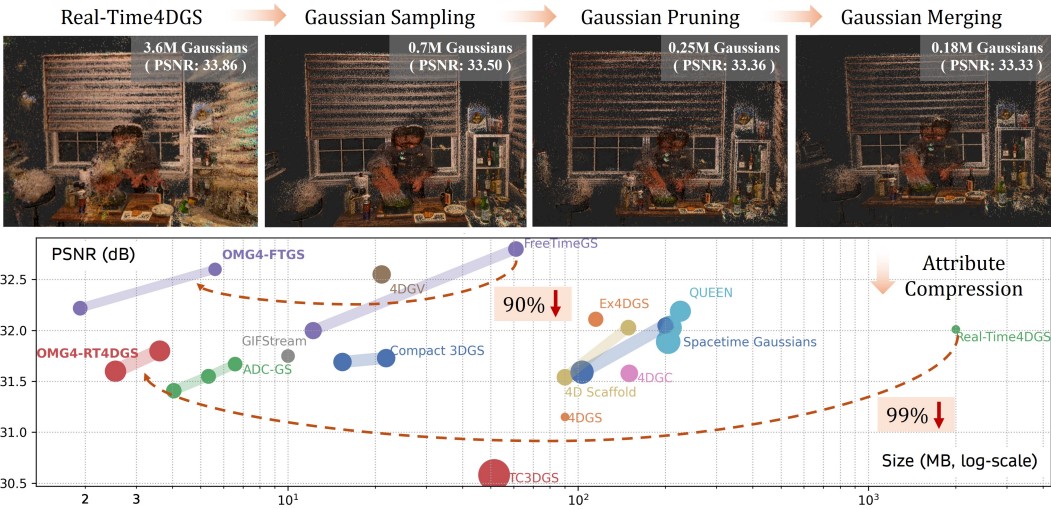

Figure 1: The overall *OMG4* pipeline and performance comparison. *OMG4* is a multi-stage 4DGS compression framework, progressively identifying important Gaussians (*Gaussian Sampling*), pruning unnecessary Gaussians (*Gaussian Pruning*), and merging similar Gaussians (*Gaussian Merging*), followed by attribute compression. The rate-distortion curve shows that *OMG4* achieved significant improvements over recent state-of-the-art methods (larger circles indicate higher FPS).

## ABSTRACT

4D Gaussian Splatting has emerged as a new paradigm for dynamic scene representation, enabling real-time rendering of scenes with complex motions. However, it faces a major challenge of storage overhead, as millions of Gaussians are required for high-fidelity reconstruction. While several studies have attempted to alleviate this memory burden, they still face limitations in compression ratio or visual quality. In this work, we present *OMG4* (Optimized Minimal 4D Gaussian Splatting), a framework that constructs a compact set of salient Gaussians capable of faithfully representing 4D Gaussian models. Our method progressively prunes Gaussians in three stages: (1) *Gaussian Sampling* to identify primitives critical to reconstruction fidelity, (2) *Gaussian Pruning* to remove redundancies, and (3) *Gaussian Merging* to fuse primitives with similar characteristics. In addition, we integrate implicit appearance compression and generalize Sub-Vector Quantization (SVQ) to 4D representations, further reducing storage while preserving quality. Extensive experiments on standard benchmark datasets demonstrate that *OMG4* significantly outperforms recent state-of-the-art methods, reducing model sizes by over 60% while maintaining reconstruction quality. These results position *OMG4* as a significant step forward in compact 4D scene representation, opening new possibilities for a wide range of applications.

## 1 INTRODUCTION

3D Gaussian Splatting (Kerbl et al., 2023) has recently achieved remarkable success, becoming a backbone for diverse 3D vision tasks ranging from 3D novel view synthesis and reconstruction to

downstream applications such as visual odometry (Keetha et al., 2024; Yan et al., 2024), 3D scene editing (Wang et al., 2025b; Lee et al., 2025a), and degradation-aware rendering (Lee et al., 2024a; Yu et al., 2024; Wan et al., 2025), to name a few. Building on this success, 4D Gaussian representations that explicitly model space and time have emerged as a new paradigm for dynamic scene reconstruction (Yang et al., 2024; Wu et al., 2024; Yang et al., 2023; Wang et al., 2025a). By augmenting each Gaussian primitive with temporal parameters, these approaches can effectively capture object motion and appearance variations over time, enabling photorealistic novel view synthesis in real time. This capability opens the door to a wide range of applications, such as free-viewpoint video (Girish et al., 2024b; Li et al., 2025; Sun et al., 2024), autonomous driving simulation Khan et al. (2024); Zhou et al. (2024), and VR/AR Pan et al. (2025); Jiang et al. (2024); Xu et al. (2023), where spatio-temporal coherence and real-time rendering are crucial.

Modeling the dynamic scenes using Gaussian primitives has evolved in two directions. Deformation-based methods employ a canonical set of 3D Gaussians and learns a deformation field that predicts per-primitive displacement and maps canonical primitives to each time step (Wu et al., 2024; Yang et al., 2023). The other approach treats the space-time as a single volume and optimizes a set of 4D Gaussian primitives, extending 3D Gaussians to the time axis for temporally varying appearance (Yang et al., 2024). By encoding motions within primitives rather than through warping, it can naturally handle complex non-rigid dynamics and occlusions, yielding higher-fidelity reconstructions.

Nevertheless, current 4D Gaussian representations often carry a substantial computational cost and memory footprint. The number of primitives can grow to millions (e.g., Real-Time4DGS (Yang et al., 2024) produces millions of 4D Gaussians, consuming over a gigabyte of memory), with each primitive carrying high-dimensional attributes that evolve over time. As a result, storage requirements frequently exceed practical limits, particularly under real-time constraints, on mobile devices, or in streaming scenarios. This overhead further complicates various downstream tasks, highlighting the need for effective storage reduction techniques that preserve both visual fidelity and rendering speed.

Several works have attempted to alleviate the significant storage requirement of explicit 4D representations (Yuan et al., 2025; Zhang et al., 2025; Li et al., 2025). 4DGS-1K (Yuan et al., 2025) presents a lifespan-based importance score to prune short-lived Gaussians, reducing the number of primitives and compressing the representation to hundreds of megabytes. GIFStream (Li et al., 2025) performs motion-aware pruning using feature streams and further mitigates storage overhead. On the other hand, Light4GS (Liu et al., 2025) leverages a deep context model, and ADC-GS (Huang et al., 2025) adopts an anchor-based structure and hierarchical approach for modeling motions at various scales to compress a deformation-based approach (e.g., (Wu et al., 2024)). Despite these advances, existing methods still require tens of megabytes to represent only a few seconds of dynamic scenes (e.g., 10 sec in N3DV (Li et al., 2022b)), limiting their practicality for long-duration and high-resolution dynamic contents.

In this paper, we propose *OMG4* (Optimized Minimal 4D Gaussian Splatting), a novel framework designed to reconstruct dynamic scenes with high fidelity and compact model size. Our approach is primarily based on Real-Time4DGS (Yang et al., 2024), which represents a dynamic scene as a 4D volume parameterized by a set of millions of 4D primitives, demanding substantial storage. We introduce a multi-stage optimization pipeline that progressively reduces the number of Gaussians, consisting of *Gaussian Sampling*, *Gaussian Pruning*, and *Gaussian Merging*. Furthermore, we incorporate implicit appearance modeling and generalize the Sub-Vector Quantization (SVQ) framework (Lee et al., 2025b), originally developed for static scenes, to dynamic 4D representations for additional compression.

We begin by analyzing the spatio-temporal properties of each Gaussian through its contribution to the rendered image. This analysis motivates *Gaussian Sampling*, which employs gradient-based scores to capture the impact of Gaussians in both static and dynamic regions, retaining only the salient ones. To further refine the representation, *Gaussian Pruning* eliminates redundant Gaussians, while *Gaussian Merging* leverages inter-Gaussian correlations to identify and fuse Gaussians with similar attributes, yielding a more compact set of primitives. These steps collectively provide a compact yet expressive Gaussian representation of dynamic scenes. Once we construct a compact Gaussian set through these steps, we encode high-dimensional appearance attributes of Gaussians

using a small MLP. We subsequently apply the SVQ that we extend for 4D representation to other attributes, compressing the model size.

We conduct comprehensive experiments on the N3DV (Li et al., 2022b) and MPEG (Li et al., 2025) datasets and evaluate the proposed method under various metrics. To the best of our knowledge, we achieve state-of-the-art performance under a strict memory budget of around 3 MB, significantly reducing the volume of the baseline model by three orders of magnitude. Notably, compared to GIFStream (Li et al., 2025), a recent state-of-the-art approach, our method reduces storage size by approximately 65% (from 10.0 MB to 3.61 MB in the N3DV dataset) while maintaining comparable reconstruction quality. We believe that the proposed approach represents a promising step for the field, opening new avenues for various research directions and practical applications.

To sum up, our contributions are as follows:

- We propose a novel multi-stage framework, progressively reducing the number of Gaussians, *Gaussian Sampling*, *Gaussian Pruning*, and *Gaussian Merging*, while maintaining the reconstruction quality.

- We generalize Sub-Vector Quantization (SVQ) for 4D representations together with implicit appearance compression, enabling highly compact yet high-fidelity models.

- To the best of our knowledge, we achieve state-of-the-art performance around a 3 MB memory budget, and negligible visual quality loss compared to the baseline model while condensing the model size from gigabytes to a few megabytes.

## 2 RELATED WORK

### 2.1 3D GAUSSIAN SPLATTING AND COMPRESSION

3D Gaussian Splatting (3DGS) (Kerbl et al., 2023) enables high-fidelity scene reconstruction with real-time rendering, using 3D Gaussians as fundamental primitives. However, it typically requires millions of primitives, which incurs substantial memory costs, urging the need for effective compression solutions. Several studies (Fan et al., 2024; Girish et al., 2024a; Shin et al., 2025; Lee et al., 2024c; 2025b) have proposed primitive pruning and attribute compression. LocoGS (Shin et al., 2025) proposes a locality-aware compact representation that encodes locally coherent Gaussian attributes with a multi-scale hash grid. Compact 3DGS (Lee et al., 2024c) adopts a learnable mask to remove unnecessary Gaussians and vector quantization (VQ) to condense geometry attributes. OMG (Lee et al., 2025b) further aims for a more compact representation, introducing sub-vector quantization (SVQ) that splits the vectors into multiple small sub-vectors and applies VQ, achieving significant performance improvement. Other works, on the other hand, present anchor-based approaches to address Gaussian redundancy (Lu et al., 2023; Chen et al., 2024). Scaffold-GS (Lu et al., 2023) organizes local Gaussians around the learned anchor points and predicts their attributes with lightweight MLPs, while HAC (Chen et al., 2024) leverages a hash grid to capture spatial consistencies among the anchors. Although these approaches effectively reduce the storage requirements of 3DGS, they do not straightforwardly generalize to 4D representations.

### 2.2 DYNAMIC 3D GAUSSIAN SPLATTING AND COMPRESSION

Early efforts to extend static scene reconstruction to dynamic settings (Cao & Johnson, 2023; Fridovich-Keil et al., 2023; Wang et al., 2023; Pumarola et al., 2020; Li et al., 2022a) were primarily based on neural volumetric rendering, but suffered from high computational costs. More recently, many studies have sought to extend 3DGS to dynamic scenes, which can be broadly categorized into two approaches: (1) representing dynamic scenes with 4D Gaussian primitives that jointly encode spatial and temporal dimensions (Yang et al., 2024; Li et al., 2024), or (2) deforming 3D Gaussians at each timestamp via a deformation field (Kratimenos et al., 2024; Bae et al., 2024; Wu et al., 2024; Yang et al., 2023). Among them, Real-Time4DGS (Yang et al., 2024) achieves high-fidelity modeling of dynamic scenes by parameterizing the 4D volume with a set of 4D Gaussians. Most recently, FreeTimeGS (Wang et al., 2025a) has shown promising performance by moving 3D Gaussians over time, leveraging motion vectors.

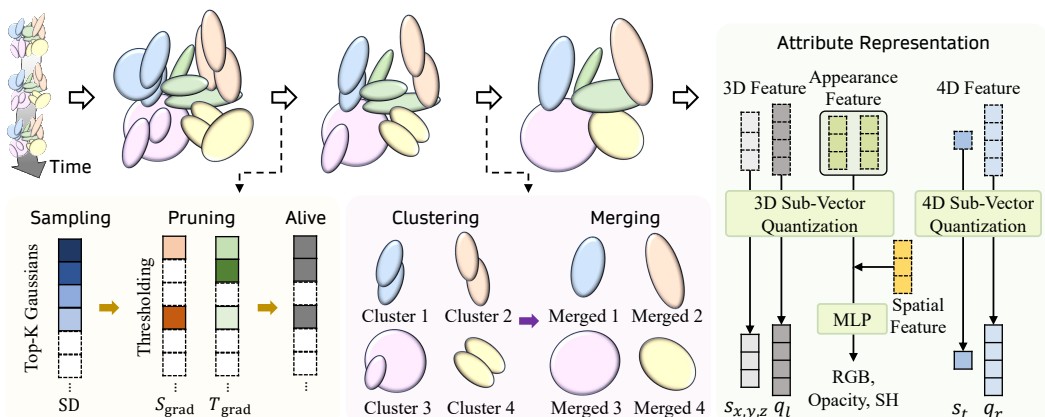

Figure 2: The overall architecture of the proposed *OMG4*.

Similar to 3DGS, dynamic extensions also require a significant number of primitives, motivating the development of lightweight methods. Within the first category, which employs 4D Gaussian primitives, CSTG (Lee et al., 2024b), MEGA (Zhang et al., 2025), and 4DGS-1K (Yuan et al., 2025) have been proposed. Among them, 4DGS-1K (Yuan et al., 2025) reduces storage by discarding Gaussians with short lifespans, achieving a model size of around 50 MB after post-processing. In contrast, other works (Liu et al., 2025; Huang et al., 2025; Chen et al., 2025; Li et al., 2025) focus on compressing deformable 3DGS. In this paper, we primarily target Real-Time4DGS (Yang et al., 2024), which demands an extremely large number of primitives (approximately 2 GB per scene), and aim to drastically reduce its memory footprint while preserving its strength in photorealistic dynamic scene reconstruction.

## 3 PRELIMINARY

Our framework is primarily built upon Real-Time4DGS (Yang et al., 2024)[1], which treats a dynamic scene as a 4D volume, parameterized with a set of 4D Gaussian primitives defined by a 4D mean at spatio-temporal space, an anisotropic 4D covariance, an opacity, and spherical harmonics (SH) coefficients. During rendering, 4D Gaussians are conditioned at timestamp $t$, yielding 3D position and covariance. For further details, please refer to the original paper.

We denote the set of Gaussians of pretrained Real-Time4DGS model by $\mathcal{P} = \{(x_i, f_i, a_i)\}_{i=1}^N$, where $x_i$ is a spatial mean (i.e., $\mu_{1:3}$), $f_i \in \mathbb{R}^3$ is the zero-th order of the SH coefficient, and $a_i$ denotes the remaining per-primitive attributes such as opacity, scale, and rotation.

## 4 METHODS

We propose a novel framework for high-fidelity dynamic scene representation with a minimal number of Gaussians consisting of Gaussian Sampling (Sec. 4.1), Gaussian Pruning (Sec. 4.2), Gaussian Merging (Sec. 4.3), and Gaussian attributes compression (Sec. 4.4). An overview of the entire pipeline is provided in Fig. 2.

### 4.1 GAUSSIAN SAMPLING

Prior methods typically rely on an excessive number of Gaussians for high-quality dynamic scene reconstruction, incurring significant storage overhead. Although recent studies observed that many Gaussians contribute only marginally to reconstruction quality, this challenge has yet to be fully addressed. In this section, we propose the Static–Dynamic Score (SD-Score), which combines a Static Score and a Dynamic Score to quantify each Gaussian's contribution. Static regions are often characterized by temporally persistent and spatially dispersed Gaussians, whereas dynamic regions tend to contain temporally short-lived and spatially concentrated Gaussians. Exploiting

---
[1]We also applied *OMG* to the recently proposed FreeTimeGS (Wang et al., 2025a). Due to space limits, we are unable to provide the details and kindly refer to the original paper for a full description.

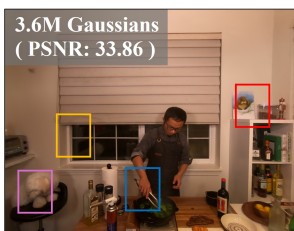 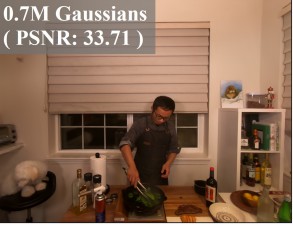 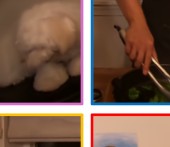 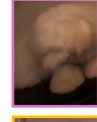 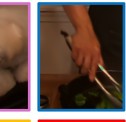 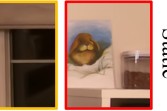

| 3.6M Gaussians ( PSNR: 33.86 ) | 0.7M Gaussians ( PSNR: 33.71 ) | | |

Dynamic

Static

| Real-Time 4DGS | Gaussian Sampling | Real-Time 4DGS | Gaussian Sampling |

Figure 3: A comparison of rendered images. (Left) Real-Time4DGS (Yang et al., 2024) with 3.6M Gaussians. (Right) 1K optimization after Gaussian Sampling with 0.7M Gaussians.

these properties, our method identifies the salient Gaussians, as shown in Fig. 3, retaining only 20% of those from a pre-trained Real-Time4DGS model (Yang et al., 2024) is sufficient to reconstruct the scene with negligible quality loss.

**SD Score.** The proposed Static–Dynamic Score (SD-Score) to calculate the importance of each Gaussian is defined as follows:

$$\mathrm{SD}^{(i)} = S_{\mathrm{grad}}^{(i)} \cdot T_{\mathrm{grad}}^{(i)}, \quad S_{\mathrm{grad}}^{(i)} = \sum_{j=1}^{N} ||\nabla_{u_{i,j}} L_j||_2, \quad T_{\mathrm{grad}}^{(i)} = \sum_{j=1}^{N} \nabla_{t_i} L_j, \tag{1}$$

where $N$ is the number of input images (# of input views $\times$ # of frames), $i$ is the Gaussian index, $L_j$ is the reconstruction loss of the input image $j$, $u_{i,j} \in \mathbb{R}^2$ denotes the projected 2D coordinate of the $i$-th Gaussian for the $j$-th image, and $t_i \in \mathbb{R}$ is the time coordinate of $i$-th Gaussian.

**Static Score.** The static score for the $i$-th Gaussian, $S_{\mathrm{grad}}^{(i)}$, is defined as an accumulation of the view-space gradients across all timesteps, capturing the overall rendering sensitivity with respect to the Gaussian's projected coordinates. In static regions, Gaussians are temporally long-lived yet relatively sparse (i.e., fewer Gaussians per unit area), so even small positional perturbations can influence the rendering loss across many timesteps, yielding higher static scores. In contrast, Gaussians associated with dynamic regions are often active only at specific time steps and are spatially dense (e.g., a larger number of Gaussians per unit area). Consequently, the effect of a single Gaussian's positional change on the overall reconstruction loss is diluted, resulting in lower static scores.

**Dynamic Score.** The dynamic score for the $i$-th Gaussian, $T_{\mathrm{grad}}^{(i)}$, is a sum of time gradients that measures the sensitivity of the reconstruction loss with respect to the time coordinate of each Gaussian. Gaussians with a high dynamic Score often imply that they play a pivotal role in representing dynamic regions, as these areas are sensitive to temporal changes, capturing the motion of the objects. We accumulate the signed time gradients rather than their magnitudes to capture consistent temporal trends. This avoids assigning high scores to Gaussians with frequent sign flips (i.e., flickering), enhancing the robustness of the score.

By combining the complementary Static and Dynamic Score, the SD-Score can provide a balanced evaluation of the overall contribution of each Gaussian. We subsequently sample the Gaussians with high $SD$ values with a sampling ratio of $\tau_{GS}$, forming a set of sampled Gaussians, $\mathcal{P}_{GS}$, and optimize it for $T_{GS}$ iterations.

### 4.2 GAUSSIAN PRUNING

While the first Gaussian sampling stage with the SD-Score yields a set of critical Gaussians, $\mathcal{P}_{GS}$, it may still include redundant Gaussians. In response, we introduce a Gaussian Pruning strategy that further refines the set by eliminating superfluous Gaussians that remain after the first stage.

Fig. 4 illustrates the space defined by $S_{\mathrm{grad}}$ and $T_{\mathrm{grad}}$, where the curve $T_{\mathrm{grad}} = c/S_{\mathrm{grad}}$ denotes the selection boundary from the first Gaussian sampling stage. Within this space, we identify Gaussians that are likely redundant, and our empirical analysis shows that those with both $S_{\mathrm{grad}}^{(i)}$ and $T_{\mathrm{grad}}^{(i)}$ values being small can be safely filtered out (gray regions in Fig. 4). To this end, we apply thresholding to

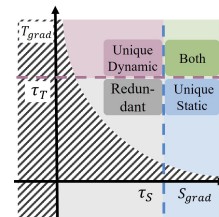 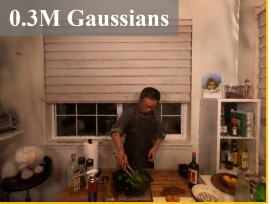 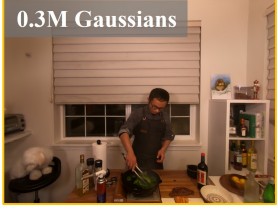 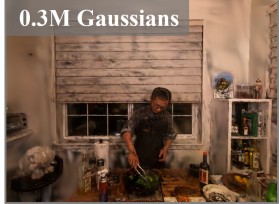

Space of Score        Unique Gaussians      Optimized Unique Gaussians      Redundant Gaussians

Figure 4: Illustration of *Gaussian Pruning*. (Left) Space defined by Static and Dynamic score and *Gaussian Sampling* Boundary. (Middle) A rendered image with unique Gaussians that satisfy both unique static and dynamic thresholds, and one after 1K optimization. (Right) A rendered image with redundant Gaussians that are not included in a unique area.

both scores, with each threshold defined as follows:

$$\mathcal{P}_{GP} = \{G_i \in \mathcal{P}_{GS} \mid (S_{\text{grad}}^{(i)} \geq \tau_S) \vee (T_{\text{grad}}^{(i)} \geq \tau_T)\}, \quad \tau_S = Q_p(\{S_{\text{grad}}^{(i)}\}), \quad \tau_T = Q_p(\{T_{\text{grad}}^{(i)}\}), \tag{2}$$

where $\tau_S$ and $\tau_T$ are the thresholds for $S_{\text{grad}}$ and $T_{\text{grad}}$ respectively, and $Q_p(\cdot)$ denotes the $p$-quantile value, reflecting the relative distribution of Gaussians. As shown in Fig. 4, the pruned set, $\mathcal{P}_{GP}$, can still represent the scene with fine details and we optimize $\mathcal{P}_{GP}$ for $T_{GP}$ iterations.

One may question the necessity of employing two distinct stages, since, in principle, sampling and pruning could be integrated into a single process. Nevertheless, our experiments indicate that separating these stages proves to be more effective, as evidenced by the ablation study in Sec. 5.2. We hypothesize that this advantage arises from the intermediate optimization performed between the two stages, which further refines the representation and ultimately results in a more compact set.

### 4.3 GAUSSIAN MERGING

The first two stages, Gaussian Sampling and Gaussian Pruning, are effective in preserving a compact and salient set of Gaussians. However, both stages evaluate the importance of each Gaussian individually, without considering similarities across Gaussians. To further reduce redundancy, we introduce a Gaussian Merging technique that clusters highly similar Gaussians and fuses them into single representative Gaussians.

**Gaussian Clustering.** We define a Similarity Score to quantify the similarity among Gaussians and identify clusters. We first divide space into a 4D grid and compute the Similarity Score among the Gaussians within the same grid cell. Such a spatio-temporal grid can group Gaussians with temporal proximity, preventing the merging of temporally mismatched Gaussians and preserving temporal coherence, while reducing computational complexity. We define the Similarity Score $S(G_i, G_j)$ between two Gaussians as a sum of spatial proximity and appearance similarity:

$$S(G_i, G_j) = -||x_i - x_j||_2^2 + \lambda ||f_i - f_j||_2^2, \tag{3}$$

where $\lambda$ is a fixed balancing weight, $x_i \in \mathbb{R}^3$ is the position, and $f_i \in \mathbb{R}^3$ is the zero-th order spherical-harmonics (RGB) coefficient of $G_i$. A higher score indicates a greater spatial-appearance similarity. For each Gaussian $G_i$, we construct a cluster of Gaussians $C_i$ by thresholding the Similarity Score with $\tau_{\text{sim}}$:

$$C_i = \{i\} \cup \{j \in \mathcal{P}_i \mid S(G_i, G_j) \geq \tau_{\text{sim}}\}, \tag{4}$$

where $\mathcal{P}_i$ denotes a set of Gaussian indices within the spatio-temporal grid cell that contain the Gaussian $G_i$. We then deduplicate identical clusters and remove subset ones, yielding a final set of maximal clusters, $\mathcal{C} = \{C_q\}_{q=1}^{N_C}$. Gaussians not included in any clusters remain as singletons.

**Gaussian Merging.** Within each cluster, we assign learnable per-Gaussian weights $w_i^x \in \mathbb{R}$ and $w_i^f \in \mathbb{R}$ to the position and appearance attributes, respectively. During training-time rendering, all Gaussians in a cluster $C_q$ are replaced by a single proxy whose elements are defined as follows:

$$\bar{x}_q = \sum_{i \in C_q} w_i^x x_i, \quad \bar{f}_q = \sum_{i \in C_q} w_i^f f_i, \quad \bar{a}_q = a_{r(C_q)}, \quad r(C_q) = \arg\max_{i \in C_q} w_i^x, \tag{5}$$

Table 1: Quantitative results on N3DV (Li et al., 2022b) dataset. All results without * mark are sourced from the original paper. † denotes post-processed models. • denotes the results excluding *Coffee Martini* scene.

| Method | PSNR↑ | SSIM↑ | LPIPS↓ | FPS↑ | # Gaussians↓ | Storage (MB)↓ |
|---|---|---|---|---|---|---|
| 1352 × 1014 Resolution | | | | | | |
| HexPlane (Cao & Johnson, 2023) | 31.70 | 0.987 | 0.075 | 0.2 | - | 250 |
| DyNeRF (Li et al., 2022a) | 29.58 | 0.980 | 0.083 | 0.015 | - | 28 |
| Real-Time4DGS (Yang et al., 2024)* | 31.96 | 0.946 | 0.051 | 57 | 3,397,510 | 2087 |
| STG (Li et al., 2024) | 32.05 | 0.946 | 0.044 | 140 | - | 200 |
| 4DGS (Wu et al., 2024) | 31.15 | - | 0.049 | 82 | - | 18 |
| MEGA (Zhang et al., 2025) | 31.49 | - | 0.057 | 77.42 | - | 25.05 |
| ADC-GS (Huang et al., 2025)–L | 31.67 | 0.981 | 0.061 | 110 | - | 6.57 |
| ADC-GS (Huang et al., 2025)–S | 31.41 | 0.972 | 0.066 | 126 | - | 4.04 |
| GIFStream (Li et al., 2025) | 31.75 | 0.938 | 0.051 | 95 | - | 10.0 |
| CSTG (Lee et al., 2024b)† | 31.69 | 0.945 | 0.054 | 186 | - | 15.4 |
| Ours–L | 31.80 | 0.941 | 0.059 | 246 | 171,136 | 3.61 |
| Ours–S | 31.60 | 0.939 | 0.064 | 258 | 137,414 | 2.54 |
| 1024 × 768 Resolution | | | | | | |
| Real-Time4DGS (Yang et al., 2024)* | 32.21 | 0.950 | 0.040 | 96 | 2770350 | 1701 |
| 4DGS-1K (Yuan et al., 2025)† | 31.87 | 0.944 | 0.053 | 805 | 666,632 | 49.50 |
| Light4GS (Liu et al., 2025)•–L | 31.69 | - | 0.053 | 37 | - | 5.46 |
| Light4GS (Liu et al., 2025)•–S | 31.48 | - | 0.064 | 40 | - | 3.77 |
| Ours–L | 32.09 | 0.946 | 0.047 | 253 | 147,153 | 3.15 |
| Ours–S | 31.81 | 0.944 | 0.051 | 246 | 120,513 | 2.25 |
| Ours•–L | 32.80 | 0.951 | 0.044 | 259 | 139,320 | 2.99 |
| Ours•–S | 32.58 | 0.950 | 0.047 | 244 | 113,862 | 2.13 |

where the both weights $w_i^x, w_i^f$ are normalized per cluster. The position and appearance are the weighted sum of the Gaussians within the cluster, and the attributes of the representative Gaussian is used for $\bar{a}_q$. We optimize both weights $w_i^x, w_i^f$ for $T_{GM}$ iterations and repeat the Gaussian Merging $M$ times, progressively increasing the grid size. The detailed process of Gaussian Merging is provided in appendix A.3.

### 4.4 Attribute Compression

Given the compact Gaussian set, we compress attributes by adapting the OMG architecture (Lee et al., 2025b), which is initially designed for static 3DGS, to the 4DGS setting with explicit time conditioning. Concretely, we follow OMG's compression pipeline, which employs an MLP to extract spatial features and three additional small MLPs that take the spatial features and position as inputs, producing opacities, static colors, and view-dependent colors, respectively. We extend these MLPs to also take the time coordinate, enabling time-varying opacity and appearance.

Furthermore, OMG introduced Sub-Vector Quantization (SVQ), which partitions an input vector into sub-vectors and quantizes each with a small codebook, demonstrating both high efficiency and reconstruction quality. In this work, we extend SVQ to dynamic 3D scene representations. Specifically, we retain the 4D Gaussian means in full precision for stability, while applying SVQ to other attributes, including the additional rotation quaternion and temporal-axis scales introduced in Real-Time4DGS (Yang et al., 2024) to model scene motion. However, quantizing both static and dynamic attributes simultaneously leads to unstable optimization. To mitigate this, we propose a staged SVQ scheme: SVQ is first applied only to 3D attributes and optimized for $T_{3D}$ iterations, after which SVQ is activated for 4D attributes. This staged approach decouples temporal and appearance sensitivities from static components, resulting in stable optimization. Finally, we compress the quantized elements using Huffman encoding (Huffman, 1952) followed by LZMA compression (Pavlov).

## 5 Experiments and Results

### 5.1 Implementation Details

For each stage, we optimize for 1,000 iterations ($T_{GS} = T_{GP} = T_{GM} = 1,000$). Starting from a pretrained model, we first perform *Gaussian Sampling*, followed by *Gaussian Pruning* at the 1,000th

Table 2: Quantiatve results on *Bartender* scene of MPEG (Li et al., 2025) dataset. All results without * mark are sourced from the original paper.

| Method | PSNR↑ | SSIM↑ | LPIPS (VGG)↓ | FPS↑ | # Gaussians↓ | Storage (MB)↓ |
|---|---|---|---|---|---|---|
| Real-Time4DGS (Yang et al., 2024)* | 32.44 | 0.895 | 0.1579 | 115 | 2,653,870 | 1630 |
| GIFStream (Li et al., 2025)–L | 31.94 | 0.879 | 0.190 | - | - | 5.3 |
| GIFStream (Li et al., 2025)–S | 31.35 | 0.872 | 0.207 | - | - | 2.3 |
| Ours–L | 32.19 | 0.892 | 0.175 | 203 | 319,906 | 6.33 |
| Ours–S | 31.91 | 0.887 | 0.190 | 238 | 196,319 | 4.00 |

Table 3: Quantitative results on applying *OMG4* to FTGS (FreeTimeGS) (Wang et al., 2025a), evaluated on N3DV (Li et al., 2022b) dataset (*We reproduced FTGS models since the codes and pretrain models are not publicly available).

| Method | PSNR↑ | SSIM↑ | LPIPS↓ | FPS↑ | # Gaussians↓ | Storage (MB)↓ |
|---|---|---|---|---|---|---|
| FTGS–L* | 32.80 | 0.9579 | 0.0398 | 129 | 500,000 | 61.04 |
| FTGS–S* | 32.00 | 0.9504 | 0.0559 | 160 | 100,000 | 12.21 |
| Ours (FTGS–L, AC Only) | 32.59 | 0.9568 | 0.0405 | 73 | 500,000 | 9.66 |
| Ours (FTGS–S, AC Only) | 32.15 | 0.9496 | 0.0551 | 107 | 100,000 | 2.12 |
| Ours (FTGS–L) | 32.62 | 0.9562 | 0.0411 | 91 | 283,977 | 5.60 |
| Ours (FTGS–S) | 32.22 | 0.9516 | 0.0491 | 112 | 90,227 | 1.92 |

iteration. *Gaussian Merging* is then applied twice, at the $2,000^{th}$ and $3,000^{th}$ iterations, respectively. After merging, attribute compression begins. We train the MLPs at the $4,000^{th}$ iteration and keep training until the end. We start applying SVQ to 3D attributes at the $9,000^{th}$ iteration, and then SVQ to 4D attributes at the $10,000^{th}$ iteration. The sampling ratio is set to $\tau_{GS} = 0.2$, and a 0.6-quantile threshold is used for *Gaussian Pruning*. We perform *Gaussian Merging* twice, increasing the spatial grid size by a factor of 1.2, while keeping a constant temporal grid size of 2.0. We used all identical hyperparameters across the dataset. Learning rates, codebook sizes, and other hyperparameters follow Real-Time4DGS (Yang et al., 2024) and OMG (Lee et al., 2025b). All experiments are conducted on a single RTX 3090 GPU.

## 5.2 RESULTS AND ANALYSIS

We conduct experiments on N3DV (Li et al., 2022b) and *Bartender* data[2] of the MPEG (Li et al., 2025) dataset. As shown in Tab. 1 and Fig. 5, we effectively reduce the model size of the baseline Real-Time4DGS (Yang et al., 2024), from 2 GB to around 3 MB, while preserving comparable visual quality. Notably, *OMG4* can reduce the storage requirement of the original Real-Time4DGS (Yang et al., 2024) by **99%**. In addition, compared to a recent state-of-the-art method GIFStream (Li et al., 2025), *OMG4* achieved **65%** reduction in storage (from 10MB to 3.61) while even improving the PSNR (31.75 vs. 31.80).

4DGS-1K (Yuan et al., 2025) achieves the highest FPS due to its visibility mask, which identifies the visible Gaussians at a given timestamp $t$, and only those Gaussians are involved in rasterization, thereby dramatically reducing computational costs. Although we do not employ any visibility masks, as improving FPS is outside our scope, we can still improve the FPS of the baseline model by $4.31\times$, thanks to our compact representation. We additionally conduct experiments on *Bartender* data of the MPEG (Li et al., 2025) dataset, which exhibits more complex motions than the N3DV (Li et al., 2022b) dataset. Following GIFStream (Li et al., 2025), we use the first 65 frames for our experiment. *OMG4* consistently outperforms the state-of-the-art methods, efficiently reconstructing the scene with minimal storage overhead, as reported in Tab. 2.

**Application on FreeTimeGS.** We further extend *OMG4* to FreeTimeGS (Wang et al., 2025a), comprehensively evaluating the proposed method beyond Real-Time4DGS (Yang et al., 2024). We begin with the pretrained FreeTimeGS models[3] with 500K Gaussians (FTGS–L) as our backbone model and apply *OMG4*, progressively pruning Gaussians from 500K to around 280K (Ours (FTGS–L)) and 90K (Ours (FTGS–S)) primitives. To see the effectiveness of the proposed multi-stage frameworks, we also provided the results of Ours (FTGS–L, AC only) and Ours (FTGS–L, AC

---

[2]Other data from the MPEG dataset were not publicly available.

[3]We reproduced FreeTimeGS models since the codes and pre-trained models are not publicly available.

GT   Real-Time4DGS   CSTG   ADC-GS   Ours

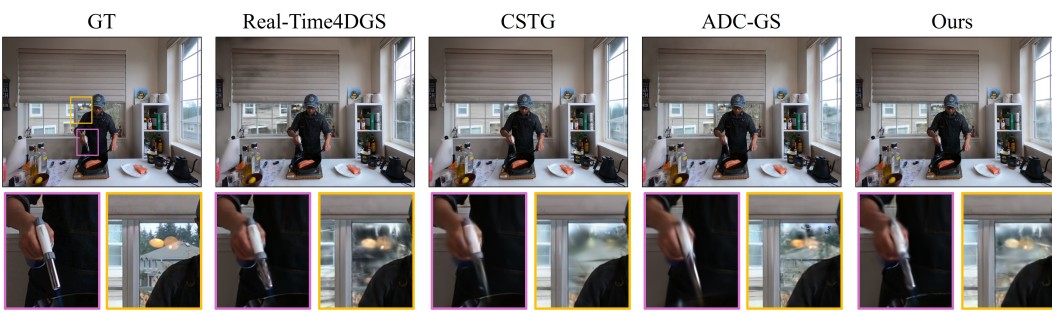

Figure 5: Qualitative results on N3DV dataset (Li et al., 2022b).

Table 4: Ablation study on each module of the proposed method where *GS*, *GP*, *GM*, *AC* refers to *Gaussian Sampling*, *Gaussian Pruning*, *Gaussian Merging* and *Attribution Compression*, respectively. *GS+GP* means applying *GS* and *GP* simutaenously at the first iteration.

| \multicolumn{5}{c}{Modules} | PSNR↑ | SSIM↑ | LPIPS↓ | FPS↑ | # Gaussians ↑ | Storage (MB)↓ |
| GS+GP | GS | GP | GM | AC | | | | | | |
| --- | --- | --- | --- | --- | --- | --- | --- | --- | --- | --- |
| | | | | | 31.96 | 0.9459 | 0.0506 | 57 | 3,397,510 | 2126 |
| | | | | ✓ | OOM | | | | | |
| | ✓ | | | | 32.07 | 0.9454 | 0.0518 | 126 | 679,502 | 13.26 |
| | ✓ | ✓ | | ✓ | 31.89 | 0.9429 | 0.0559 | 244 | 235,027 | 4.83 |
| ✓ | | | ✓ | ✓ | 31.68 | 0.9407 | 0.0606 | 217 | 171,214 | 3.61 |
| | ✓ | ✓ | ✓ | ✓ | 31.80 | 0.9414 | 0.0594 | 246 | 171,136 | 3.61 |

only), where we apply only the proposed attribute compression technique to the pretrained Free-TimeGS models (FTGS–L: 500K Gaussians, FTGS–S: 100K Gaussians). As Tab. 3 shows, *OMG4* can significantly reduce the storage of FreeTimeGS **by 90%**. Even under the stricter memory budget of around 2 MB, *OMG4* still remains effective, benefiting from its prior. This result highlights the versatility of *OMG4*, showing that it can be effectively applied to diverse 4D representation methods.

**Ablation Studies.** In this section, we present ablation studies on each module of *OMG4*, summarized in Tab. 4. First, directly applying *Attribute Compression* to Real-Time4DGS (Yang et al., 2024) incurs out-of-memory failure, due to a larger number of Gaussians. When we implement *Gaussian Sampling* only, we can remove 80% of Gaussians, even outperforming Real-Time4DGS. We assume this is because *Gaussian Sampling* can act as a regularizer by eliminating noisy Gaussians, hence raising visual quality. Adding *Gaussian Pruning* can reduce the memory footprint from gigabytes to a few megabytes, and *Gaussian Merging* achieves the minimal number of Gaussians, integrating highly correlated Gaussians. It may appear possible to perform *Gaussian Sampling* and *Gaussian Pruning* simultaneously. However, our ablation study (fifth row, *GS+GP* in Tab. 4) shows that doing so results in a PSNR drop of 0.12 dB. This finding highlights the importance of the proposed multi-stage pipeline: by first stabilizing the optimization with *Gaussian Sampling* and then progressively reducing the number of Gaussians through *Gaussian Pruning*, we are able to maintain reconstruction quality while ensuring stable training. One possible explanation is that $T_{GS}$ iteration training before *Gaussian Pruning* stabilizes the whole optimization process. Therefore, our pipeline separates *Gaussian Sampling* and *Gaussian Pruning* with a sufficient optimization term.

# 6   CONCLUSION

We introduce *OMG4*, a compact dynamic-scene representation that progressively reduces the number of primitives and compresses the Gaussian attributes. We present *Gaussian Sampling* and *Gaussian Pruning*, which reduce the number of Gaussians significantly and further leverage the correlation of Gaussians to fuse the similar Gaussians in the *Gaussian Merging* stage. Lastly, we couple these Gaussian minimization techniques with implicit appearance encoding and 4D extension of Sub-Vector Quantization (SVQ). Consequently, *OMG4* compresses Real-Time4DGS by three orders of magnitude at comparable fidelity. *OMG4* also transfers to the recent state-of-the-art method, FreeTimeGS, achieving 90% storage reduction while maintaining high reconstruction quality. With its significant performance improvement, we believe *OMG4* marks an important advance in 4D scene representation, opening new opportunities for research and various practical applications.

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

## A  APPENDIX

### A.1  USING LLM FOR POLISHING WRITING

We made limited use of a large language model (LLM) to assist with paper preparation, specifically for refining wording, improving clarity, and providing light proofreading. Its role was restricted to suggesting alternative phrasings and correcting minor grammatical issues, while all technical content and substantive writing were carried out by the authors. As this partial assistance helped improve readability and logical flow, we disclose it here in accordance with the ICLR policy.

## A.2 ADDITIONAL IMPLEMENTATION DETAILS

In this section, we describe the implementation details and experimental settings in detail. As mentioned, we follow the Sub-Vector Quantization (SVQ) setup of OMG (Lee et al., 2025b), where codebook size of scale $b_s$, rotation $b_r$, and encoded appearance feature $b_f$ are $2^9$, $2^{13}$, and $2^{10}$, respectively. Real-Time4DGS (Yang et al., 2024) introduces new attributes, $s_t$, scale along the time axis, and $q^l$, an additional quaternion to define the rotation matrix. We set their codebook size to $2^9$ and $2^{13}$, respectively. In terms of our small model, we decrease the size of the codebooks for 3D attributes to $b_s = 2^8$, $b_r = 2^{13}$, and $b_f = 2^{10}$, while keeping those for the two 4D attributes as they are. We use the same codebook setup to both N3DV (Li et al., 2022b) and MPEG (Li et al., 2025) datasets. We set $\tau_{GS} = 0.4$ and 0.6-quantile for MPEG (Li et al., 2025) dataset.

For our FreeTimeGS (Wang et al., 2025a) experiment, we use $\tau_{GS} = 0.7$ and the quantile at level 0.6 for *Gaussian Pruning*, for the large model, while $\tau_{GS}$ is set to 0.3 and a 0.4-quantile is adopted for the small model, to aggressively prune the primitives. Similar to the Real-Time4DGS (Yang et al., 2024) implementation, we also sub-vector quantize the origin attributes that FreeTimeGS introduces, velocity and duration. We keep the identical codebook size for both scales of the model, and we use the codebook size $2^{10}$ for the velocity and duration.

## A.3 ANALYSIS ON METHOD

**SD Score.** In this section, we analyze the proposed $SD$ score in *Gaussian Sampling* stage. *Gaussian Sampling* aims to extract a subset of Gaussians that can faithfully represent both static and dynamic regions. Previous work (Oh et al., 2025) has attempted to leverage the variance of time distribution, $\Sigma_t$, as a cue to detect separate static and dynamic regions. Gaussians located at the regions with various motions tend to have relatively smaller values of $\Sigma_t$ compared to those lying on the stationary area, as rapid changes in appearance or geometry restrict each primitive to have a short lifespan, being temporally localized. However, using $\Sigma_t$ alone to select a subset with a limited budget can be biased toward extremes because sampling dynamic Gaussians by taking the bottom $\Sigma_t$ subset focuses on a few high-velocity regions (e.g., moving hands in Fig. 6), while neglecting other dynamic structures such as secondary body motion. This is because $\Sigma_t$ is agnostic to the visual contribution, oversampling transients that are temporally sharp. On the other hand, our $SD$ Score addresses this by combining gradient-based contribution with temporal support. Under the same sampling budget, it can distribute the budget across multiple dynamic regions while keeping enough static structures, as shown in Fig. 6, producing better perceptual fidelity with the same number of Gaussians.

Moreover, some online 4D reconstruction approaches (Dai et al., 2025; Liu et al., 2024) adopts 2D optical flow to distinguish dynamic and static Gaussians using 2D optical flow. To further examine whether such approach can benefit our pipeline, we adopt the optical-flow–based dynamic region estimation proposed in (Liu et al., 2024) and use it as a Dynamic Score. While streaming-based methods typically compute optical flow of each streamed frame, our method requires evaluating each Gaussian's contribution to the entire scene globally. To this end, we accumulate optical-flow magnitudes across all frames. We then project each Gaussian onto the image plane and accumulate the flow magnitudes of pixels that fall within the corresponding 2D dynamic regions, thereby assigning an optical flow magnitude to each 4D Gaussian. If the accumulated optical flow exceeds a threshold $\tau_{flow}$, we mark such Gaussians as dynamic Gaussians.

The segmented results are provided in Fig. 7 and quantitative results on the N3DV dataset are delivered in Tab. 5. The optical flow-based variant performs noticeably worse than our temporal gradient-based Dynamic Score in terms of both reconstruction quality and compression efficiency. We assume this is because optical flow is highly sensitive to frame-to-frame pixel changes and thus primarily highlights a few extremely fast-moving regions, rather than Gaussians that are essential for maintaining the global scene structure. As illustrated in Fig. 7, the top 10% Gaussians ranked by optical flow predominantly capture highly dynamic parts, but fail to select representative dynamic Gaussians necessary for preserving the overall scene, similar to using $\Sigma_t$ (Oh et al., 2025). In other words, optical flow mainly measures how much a Gaussian move and does not guarantee sufficient coverage of the entire dynamic region under a tight sampling budget. In contrast, our temporal gradient-based Dynamic Score directly measures how much each Gaussian contributes to the reconstruction loss over time, leading to far more stable and reliable selections.

Table 5: Comparison of replacing the Dynamic Score with optical flow.

| Methods | PSNR↑ | SSIM↑ | LPIPS↓ | # Gaussians ↓ | Storage (MB)↓ |
|---|---|---|---|---|---|
| Optical flow | 27.19 | 0.840 | 0.156 | 150,889 | 3.20 |
| Temporal gradient (Ours) | 31.60 | 0.939 | 0.064 | 137,414 | 2.54 |

**Gaussian Merging.** Algorithm. 1 describes a detailed process of *Gaussian Merging*. As mentioned earlier, our *Gaussian Merging* performs correlation-considered Gaussian removal based on the locality of Gaussians, while *Gaussian Sampling* and *Gaussian Pruning* rather aim to identify the salience or contribution of a single Gaussian.

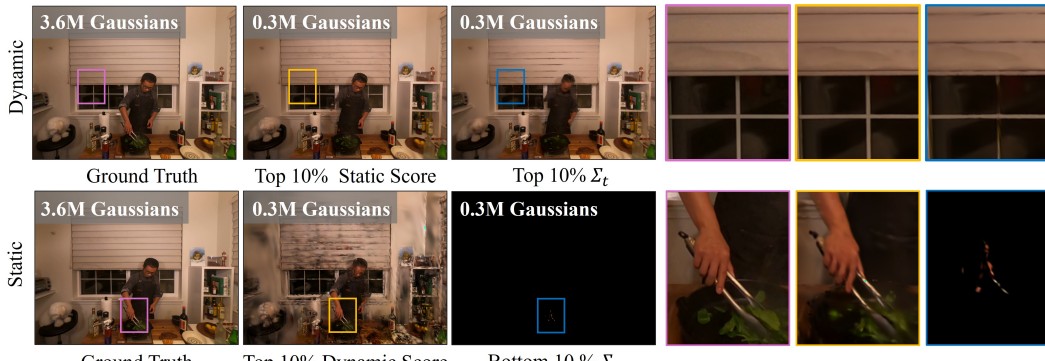

Figure 6: Effect of the sampling criterion under a fixed sampling budget. (Left) Rendered images with the full model. (Middle): Our $SD$ Score selection using top 10% Static score and Dynamic Score. (Right) $\Sigma_t$-based sampling with the same sampling ratio.

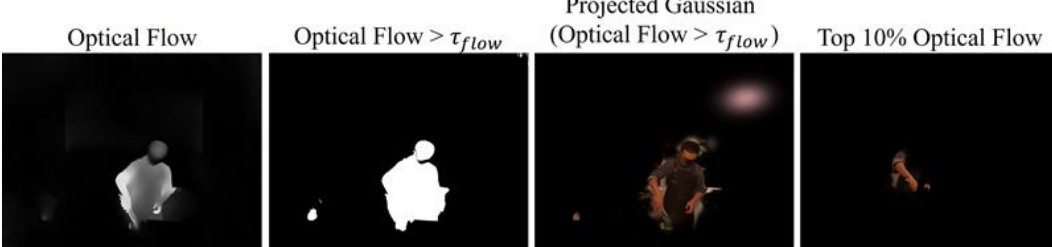

Figure 7: Visualization of 2D optical flow-based Dynamic score. (Left) Predicted 2D optical flow. (Middle-Left): Optical flow thresholded with $\tau_{flow}$. (Middle-Right) Rendered image with the Gaussians projected on high-optical flow region. (Right): Rendered image with the Gaussians of top 10% Dynamic score.

**Ablation Studies.** We further conduct ablation studies on the proposed method, evaluated on *cook spinach* and *sear steak* data of N3DV dataset (Li et al., 2022b). We compare each hyperparameter of the Gaussian removal stage. Tab. 6 and Tab. 7 show the comparison on different pruning ratio $p$ and the number of *Gaussian Merging* implementation, respectively. As expected for Gaussian-based models, the reconstruction quality increases monotonically with the number of Gaussians. We experimentally set these hyperparameters, considering the storage and image fidelity tradeoff.

### A.4 PER-SCENE RESULTS

In this section, we provide per-scene results of N3DV dataset (Li et al., 2022b) in Tab. 8

**Algorithm 1** Gaussian Merging

**Require:** Gaussian set $\mathcal{P}_{GP}$; clusters $\mathcal{C} = \{C_q\}_{q=1}^{N_C}$; training steps $T_M$

   Initialize per-member logits $\ell_i^x \leftarrow 0$, $\ell_i^f \leftarrow 0$ for all $i$

   **for** $t = 1$ **to** $T_M$ **do**

      **for all** clusters $C_q$ **do**

         $w_i^x \leftarrow \sigma(\ell_i^x) \big/ \sum_{j \in C_q} \sigma(\ell_j^x)$    for $i \in C_q$               ▷ Position weights

         $w_i^f \leftarrow \sigma(\ell_i^f) \big/ \sum_{j \in C_q} \sigma(\ell_j^f)$    for $i \in C_q$              ▷ Appearance weights

         $\bar{x}_q \leftarrow \sum_{i \in C_q} w_i^x x_i$;   $\bar{f}_q \leftarrow \sum_{i \in C_q} w_i^f f_i$

         $\bar{a}_q \leftarrow a_{r(C_q)}$,    $r(C_q) = \arg\max_{i \in C_q} w_i^x$

      **end for**

      $\mathcal{P}_{\text{proxy}} \leftarrow \{(\bar{x}_q, \bar{f}_q, \bar{a}_q)\}_{q=1}^{N_C} \cup \{(x_i, f_i, a_i) : i \notin \bigcup_q C_q\}$

      Render $\mathcal{P}_{\text{proxy}}$, compute loss $\mathcal{L}$, and update $\{\ell_i^x, \ell_i^f\}$ by backprop

   **end for**

   **return** pruned set $\mathcal{P}_{\text{proxy}}$

Table 6: Ablation study on the quantile level of $p$ of *Gaussian Pruning*.

| Methods | PSNR ↑ | SSIM ↑ | LPIPS ↓ | # Gaussians ↓ | Storage (MB) ↓ |
|---|---|---|---|---|---|
| $p = 0.6$ | 33.51 | 0.956 | 0.045 | 225,286 | 4.65 |
| $p = 0.7$ | 33.47 | 0.956 | 0.046 | 172,193 | 3.95 |
| $p = 0.8$ | 33.40 | 0.955 | 0.048 | 143,030 | 3.07 |
| $p = 0.9$ | 33.05 | 0.950 | 0.055 | 83,052 | 1.91 |

Table 7: Ablation study on the number of *Gaussian Merging* execution $M$.

| Methods | PSNR ↑ | SSIM ↑ | LPIPS ↓ | # Gaussians ↓ | Storage (MB) ↓ |
|---|---|---|---|---|---|
| $M = 1$ | 33.44 | 0.955 | 0.047 | 161663 | 3.43 |
| $M = 2$ | 33.40 | 0.955 | 0.048 | 143030 | 3.07 |
| $M = 3$ | 33.32 | 0.954 | 0.049 | 130320 | 2.82 |
| $M = 4$ | 33.26 | 0.954 | 0.0501 | 120347 | 2.63 |

Table 8: Quantitative results on N3DV (Li et al., 2022b) dataset. All results without * mark are sourced from the original paper. † denotes post-processed models.

| Method | Coffee Martini | Cook Spinach | Cut Roasted Beef | Flame Salmon | Flame Steak | Sear Steak | Average |
|---|---|---|---|---|---|---|---|
| PSNR↑ | | | | | | | |
| HexPlane (Cao & Johnson, 2023) | - | 32.04 | 32.54 | 29.47 | 32.08 | 32.38 | - |
| Real-Time4DGS (Yang et al., 2024)* | 28.33 | 32.93 | 33.85 | 29.38 | 33.51 | 33.51 | 32.01 |
| STG (Li et al., 2024) | 28.61 | 33.18 | 33.52 | 29.48 | 33.64 | 33.89 | 32.05 |
| 4DGS (Wu et al., 2024) | 27.34 | 32.46 | 32.90 | 29.20 | 32.51 | 32.49 | 31.15 |
| MEGA (Zhang et al., 2025) | 27.84 | 33.08 | 33.58 | 28.48 | 32.27 | 33.67 | 31.49 |
| ADC-GS (Huang et al., 2025)-L | - | 32.34 | 31.88 | 29.01 | 32.65 | 32.48 | 31.67 |
| GIFStream (Li et al., 2025) | 28.14 | 33.03 | 33.19 | 28.51 | 33.76 | 33.83 | 31.75 |
| Ours–L | 28.10 | 33.12 | 33.61 | 28.70 | 33.57 | 33.69 | 31.80 |
| Ours–S | 27.98 | 32.93 | 33.30 | 28.40 | 33.42 | 33.56 | 31.60 |
| SSIM↑ | | | | | | | |
| HexPlane (Cao & Johnson, 2023) | - | - | - | - | - | - | - |
| Real-Time4DGS (Yang et al., 2024)* | - | - | - | - | - | - | - |
| STG (Li et al., 2024) | - | - | - | - | - | - | - |
| 4DGS (Wu et al., 2024) | 0.905 | 0.949 | 0.957 | 0.917 | 0.954 | 0.957 | 0.939 |
| MEGA (Zhang et al., 2025) | - | - | - | - | - | - | - |
| ADC-GS (Huang et al., 2025) | - | - | - | - | - | - | - |
| GIFStream (Li et al., 2025) | 0.905 | 0.950 | 0.947 | 0.916 | 0.957 | 0.958 | 0.938 |
| Ours–L | 0.910 | 0.952 | 0.954 | 0.916 | 0.958 | 0.958 | 0.941 |
| Ours–S | 0.907 | 0.950 | 0.950 | 0.086 | 0.956 | 0.956 | 0.939 |
| LPIPS↓ | | | | | | | |
| HexPlane (Cao & Johnson, 2023) | - | 0.082 | 0.080 | 0.078 | 0.066 | 0.070 | |
| Real-Time4DGS (Yang et al., 2024)* | - | - | - | - | - | - | - |
| STG (Li et al., 2024) | 0.069 | 0.037 | 0.036 | 0.063 | 0.029 | 0.030 | 0.044 |
| 4DGS (Wu et al., 2024) | - | - | - | - | - | - | - |
| MEGA (Zhang et al., 2025) | 0.077 | 0.047 | 0.048 | 0.073 | 0.053 | 0.040 | 0.056 |
| ADC-GS (Huang et al., 2025) | - | - | - | - | - | - | - |
| GIFStream (Li et al., 2025) | - | - | - | - | - | - | - |
| Ours–L | 0.085 | 0.049 | 0.049 | 0.080 | 0.458 | 0.458 | 0.059 |
| Ours–S | 0.091 | 0.054 | 0.055 | 0.086 | 0.049 | 0.049 | 0.064 |

