# OpenReview forum: "Optimized Minimal 4D Gaussian Splatting"
_ICLR.cc/2026/Conference — Submitted to ICLR 2026_

### Official Review · Reviewer_3wjx · 2025-10-25

**Soundness:** 3
**Presentation:** 3
**Contribution:** 2
**Rating:** 4
**Confidence:** 4

**Summary:**

This paper proposes a pipeline to compress Gaussian Splatting based 4D dynamic spatial representations. The method concludes several stages, as selecting salient Gaussians based on Static-Dynamic scores, pruning low-importance primitives, merging similar Gaussians within grids, and compressing attributes with MLPs and arithmetic compressors. The resulting 4DGS can provide a highly compact storage size while maintaining the visual quality. This pipeline is applicable to multiple off-the-shelf baselines and achieves compactness.

**Strengths:**

1. One impressive point of this work is that this paper proposes a universally applicable pipeline as a post-processing for multiple 4DGS representations. This highlights the method's generality, which I think is meaningful for this task.
2. The SD-Score, presenting the spatial and temporal sensitivity, guides the pipeline to distinguish the important Gaussians and provides effective compression.
3. The performances are impressive compared to the baselines, achieving a low storage size of several MB to represent a 4D dynamic space.

**Weaknesses:**

1. Although the proposed method provides superior compactness, the proposed components in the method are not very impressive in the aspect of novelty.  The Gaussian importance score, MLP based color coding, and post-processing compressors are upgraded from 3DGS compression techniques or previously established in other 4D Gaussian compression works. These make the novelty of this work weaker, and the authors fail to distinguish their designs from the previous techniques. My own opinion is that the author can emphasize more on the generality of the proposed method: The authors can establish the whole pipeline as a more robust and general post-processing progress to all (or most of) the previous 4DGS baselines, and investigate how to specifically implement the proposed method to each distinct baselines and fit the special property of each method.
2. Another concern is a highly engineered pipeline with many hyperparameters. The whole pipeline includes multiple thresholds, quantiles, grid sizes and merge / iteration budgets. These elements raise concerns on the robustness of the proposed method. Naturally, some scenes require high quantity of Gaussians for representation, while some other may not. Some scenes contain complex lattice or textures. The discussion or ablation on these issues should be provided for a solid presentation.

**Questions:**

Regarding the weaknesses listed above:
1. The authors are suggested to provide more clarifications on the generality of the proposed method, such as elaborating on how the proposed pipeline fits different baselines and analysis on the effectiveness on them. The authors are recommended to provide further attempts on other baselines and clarify the effectiveness, and in which cases the method fits better.
2. The authors are recommended to provide more justifications and ablations on the hyperparameters, and how these hyperparameters are affecting the final compression performance. Are these hyperparameters tuned for each scene or each dataset?
3. Some previous 4D reconstruction methods use zero-shot dynamic-static distinguishment [R1, R2]. Can they benefit the proposed pipeline? These methods should also be considered or discussed.

[R1] Dai, P., Zhang, P., Dong, Z., Xu, K., Peng, Y., Ding, D., Shen, Y., Yang, Y., Liu, X., Lau, R.W. and Xu, W., 2025. 4d gaussian videos with motion layering. ACM Transactions on Graphics (TOG), 44(4), pp.1-14.

[R2] Liu, Z., Hu, Y., Zhang, X., Song, R., Shao, J., Lin, Z. and Zhang, J., 2024. Dynamics-Aware Gaussian Splatting Streaming Towards Fast On-the-Fly 4D Reconstruction. arXiv preprint arXiv:2411.14847.

I am looking forward to the authors' reply on the above issues, based on which I am willing to further adjust my review.

---

> ### Author Response · Authors · 2025-11-21
>
> ## **Generality of the method**
> We applied the proposed method on two baselines, RT4DGS and FTGS, and showed the effectiveness and generality of OMG4 on different models. We further evaluate our method to STG, and the corresponding results are provided in "Generality of the proposed method" part of General Response section.
>
> ## **Hyperparameters**
> First, we did not fine-tune the hyperparameters for each scene, but kept the same hyperparameters across all scenes in each dataset (i.e., 6 scenes in N3DV dataset were trained with the identical hyperparameters). Although using multiple hyperparameters could be seen as a potential drawback, it also offers a clear practical benefit: it allows us to easily control the rate–distortion (RD) trade-off and FPS. The table below illustrates how adjusting these hyperparameters shifts the RD curve. Ours-L, Ours-M, and Ours-S are obtained simply by changing the sampling ratio $\tau_{GS}$ and the pruning ratio $p$ . Ours-T further reduces the model size by additionally decreasing the codebook size. This demonstrates that our framework provides a simple and interpretable way to navigate the RD trade-off using only a few hyperparameters, enabling OMG4 to be readily adapted to different storage and quality budgets without modifying the underlying backbone.
>
> | Method  | PSNR ↑ | SSIM ↑ | LPIPS ↓ | FPS ↑ | # Gaussians ↓ | Storage (MB) ↓ |
> |---------|--------|--------|---------|-------|----------------|----------------|
> | Ours–L  | 31.99  | 0.943  | 0.056   | 202   | 283,297 | 5.75  |
> | Ours–M  | 31.80  | 0.941  | 0.059   | 246   | 171,136 | 3.61 |
> | Ours–S  | 31.60  | 0.939  | 0.064   | 258   | 137,414 | 2.54 |
> | Ours–T  | 31.47  | 0.937  | 0.067   | 275   | 111,402 | 2.09 |
>
> As noted, some scenes require more Gaussians, while others can be represented more compactly. We explicitly test OMG4 on a broad spectrum of scenarios: from standard N3DV dataset to the more complex MeetRoom dataset, and to highly dynamic sequences such as SA-VVC and Bartender scene. Across all these scenes, using a single  configuration of OMG4 (A setting for “Ours -M” in above table), we consistently obtain large compression ratios (up to 90–95\% model size reduction) with comparable visual quality. We believe the new experiments on various dataset provided in “Experiments on additional dataset” part of General Response section provide empirical evidence that OMG4 is robust and can handle scenes with varying complexity without delicate per-scene tuning.
>
> ## **Zero-shot dynamic-static distinguishment**
> To examine whether zero-shot dynamic-static Gaussian distinction approaches could benefit our pipeline, we leverage the optical-flow–based dynamic region estimation proposed in [R2] and use it as a replacement for our Dynamic Score.
> While prior streaming-based methods compute optical flow on a per-frame basis as the sequence progresses, our method requires evaluating each Gaussian’s contribution to the entire frames. Therefore, we accumulated optical-flow magnitudes across all frames. To assign 2D flow magnitude to 4D Gaussians, we projected each Gaussian onto the image plane and accumulated magnitudes for Gaussians that fell within the 2D dynamic regions. If the accumulated optical flow exceeds a threshold $\tau_{flow}$, we marked such Gaussians as dynamic Gaussians. The segmented results are provided in Fig. 7 in the Appendix, and quantitative results on N3DV dataset are reported in the table below. The qualitative results are provided in the [link](https://anonymous.4open.science/r/OMG4_Qualitative).
>
> | Method                    | PSNR ↑ | SSIM ↑  | LPIPS ↓ | # Gaussians ↓ | Storage (MB) ↓ |
> |---------------|--------|---------|---------|----------------|----------------|
> | Optical flow              | 27.19  | 0.840  | 0.156  | 150,889 | 3.20 |
> | Temporal gradient (Ours)  | 31.60  | 0.939   | 0.064   | 137,414 | 2.54 |
>
> The optical flow-based approach noticeably performs worse than our method. We assume this is because optical flow is highly sensitive to frame-to-frame pixel changes. As a result, it highlights only a few extremely fast-moving regions rather than Gaussians essential for maintaining the global scene structure. As shown in Fig. 7 in Appendix, the top 10\% Gaussians ranked by optical flow only capture highly dynamic parts (rightmost image in the figure), but they fail to select representative dynamic Gaussians necessary for preserving the overall scene.
> Since the optical flow focuses on “how much a Gaussian move,” it does not guarantee sufficient coverage of the entire dynamic region. Meanwhile, our temporal gradient-based Dynamic Score directly measures how much each Gaussian contributes to the reconstruction loss, yielding far more stable and reliable results. Therefore, although the zero-shot approaches in [R1] and [R2] are interesting directions, we found that they do not align with our goal of selecting Gaussians based on their true contribution to high-quality scene reconstruction.

---

> > ### Comment · Reviewer_3wjx · 2025-11-27
> > **Thanks for the rebuttal.**
> >
> > I appreciate the rebuttal provided by the authors. My initially raised concerns are solved and I will adjust my score.

---

> > > ### Author Response · Authors · 2025-11-30
> > >
> > > We sincerely thank the reviewer for the constructive feedback, which helped us further improve the paper, and we appreciate that you reconsidered your evaluation and raised the score.

---

### Official Review · Reviewer_jaTu · 2025-10-27

**Soundness:** 2
**Presentation:** 3
**Contribution:** 3
**Rating:** 6
**Confidence:** 5

**Summary:**

OMG4 (Optimizing Minimal 4D Gaussian Distributions) constructs a compact set of salient Gaussians that faithfully represent the 4D Gaussian model.
The method progressively prunes Gaussians in three stages:
(1) **Gaussian sampling** to identify primitives critical to reconstruction fidelity,
(2) **Gaussian pruning** to remove redundancies, and
(3) **Gaussian merging** to fuse primitives with similar characteristics.

**Strengths:**

1. The three strategies proposed in the paper are interesting, and the writing is well-organized.
2. The proposed method achieves a significant reduction in storage while maintaining comparable performance.
3. The paper provides extensive visualization videos.

**Weaknesses:**

1. The experiments are too limited — most evaluations are conducted only on the N3DV dataset. Testing on large-scale dynamic scenes such as NVIDIA[1], Dynamic3DGS[2], or VRU [3] would make the results much more convincing.
2. The proposed compression and pruning strategies are all based on 4DGS; however, 4DGS has inherent limitations, such as its inability to handle fast motion or long-sequence dynamic videos. Therefore, this method is somewhat restricted in its applicability.

If the authors can demonstrate strong experimental results (both quantitative and qualitative) on large-scale datasets, this paper would deserve a score of 8 and be worth accepting.

[1] Neural Trajectory Fields for Dynamic Novel View Synthesis

[2] Dynamic 3D Gaussians: Tracking by Persistent Dynamic View Synthesis

[3] Swift4D: Adaptive divide-and-conquer Gaussian Splatting for compact and efficient reconstruction of dynamic scene

**Questions:**

1. How long does the model take to train?
2. Is the model initialized using Gaussian points from all frames to represent the entire scene, followed by the sampling, merging, and pruning strategies? If so, wouldn’t this result in high GPU consumption and require long training times — possibly several hours?

---

> ### Author Response · Authors · 2025-11-21
>
> ## **Experiments on additional dataset**
> In the “Experiments on additional dataset” part of General Response section, we report the results of additional experiments. Since the VRU dataset is not publicly accessible, we exclude it from our evaluation and instead conduct experiments on the MeetRoom dataset used in [3], as well as three other datasets that exhibit complex motions and long frame sequences.
>
> ## **Experiments on other baselines**
> We agree that RT4DGS has several limitations, and for this reason we did not rely solely on RT4DGS. In the main paper, we conducted experiments on FTGS as well, and observed strong performance (Please refer to Tab. 3 in the main paper). FTGS is better suited than RT4DGS for handling complex motions and longer sequences, which allowed it to achieve better results overall. In particular, on challenging datasets such as SA-VVC, where dancing and other highly non-rigid motions are prevalent, RT4DGS failed to train successfully, whereas FTGS produced stable reconstructions. Applying OMG4 to FTGS enabled effective compression without sacrificing noticeable visual quality. Please also refer to the qualitative results provided at the attached link for visual comparisons on these challenging sequences. In addition to these existing results, we have now further evaluated OMG4 on STG. The corresponding results for STG are reported in the “Generality of the proposed method” part of the General Response section.
>
> ## **Training time**
> We reported the detailed training time of each module measured on N3DV dataset in “Training time analysis” part in General Response section. OMG4 takes 16.35 minutes for preprocessing (computing view-space and temporal gradient) and 28.83 minutes for optimization. Also, as you pointed out, because OMG4 is a post-processing step to a pretrained model, it holds information from all frames. However, at the very beginning of OMG4 training, we apply Gaussian Sampling to aggressively reduce the number of primitives (e.g., sampling only 20\% of the Gaussians on the N3DV dataset). This substantial reduction in Gaussian count allows us to speed up optimization by roughly 2× compared to the pretrained model, reducing the per-iteration time from 0.31 s/iter to 0.148 s/iter.

---

> ### Comment · Reviewer_jaTu · 2025-11-22
>
> Thank you for the authors’ detailed response. However, some of my original concerns still remain unresolved. My specific questions are as follows:
>
> 1. I am not fully convinced by the authors’ characterization of the MeetRoom dataset as a complex motion dataset. To my understanding, it represents a relatively small meeting-room scenario with limited motion complexity, and it does not constitute a long-frame sequence, as each scene contains only around 300 frames. For many existing methods, performance on this dataset is already close to overfitting. Therefore, evaluating OMG4 solely on this small-motion dataset may not sufficiently demonstrate the generalization ability of the proposed method. Although OMG4 is designed for compression, the underlying methods it builds upon—RT4DGS and FTGS—have known limitations, and OMG4 may inherit or even amplify these constraints.
>
> 2. Based on the experiments presented by the authors (Experiments on additional dataset), performance degradation is observed on Bartender (–0.3 to –1.6 dB), Flame Salmon (–0.4 to –1 dB), and the VVC dataset (–2 dB). The drop appears more noticeable when compared with the original FTGS, suggesting that **OMG4 may not achieve compression while preserving reconstruction quality, which is a key goal for compression methods.**
>
> I sincerely appreciate the authors’ efforts and hope that further clarification can help address these concerns.

---

> ### Author Response · Authors · 2025-11-30
>
> We sincerely thank the reviewer for the detailed comments and suggestions, which helped us further clarify and strengthen the paper. Below, we provide our responses to the raised concerns.
>
> (1) MPEG dataset includes more challenging motions such as bottle-throwing, shaking, and dancing. Please visit the **[link](https://anonymous.4open.science/r/OMG4_Qualitative)** to see these challenging motion patterns. To further validate OMG4’s robustness under such fast non-rigid motion, we additionally evaluate on the Cinema scene from MPEG. The results are summarized in the table below.
>
> As the reviewer correctly pointed out, OMG4 inherits the behavior of the underlying pretrained backbone. Nevertheless, across three baselines — RT4DGS, FTGS, and STG — we consistently observe that OMG4 achieves strong compression with only minor visual degradation, indicating that OMG4 will readily generalize to future, stronger dynamic-scene models.
>
> In addition, we conducted experiments on GIFStream, the previous SOTA compression approach. As the results show, OMG4 outperforms GIFStream with a noticeable margin, confirming that OMG4’s improvements on challenging dynamic scenes stem from its selection and compression design rather than the choice of backbone.
>
> (2) Initially, we conducted all FTGS + OMG4 experiments with 5,000 iterations of MLP optimization in the Attribute Compression stage. However, we later found that the MLP had not fully converged for FTGS if the dataset shows complex motion or contains long sequences. Therefore, only for the FTGS + OMG4 case, we increased the number of MLP training iterations from 5,000 to 12,000. The effect of this adjustment is summarized in the table below (updated rows are written in **bold text**).
>
> For the SA-VVC dataset, we used the pretrained model provided by the SA-VVC organizers and then applied OMG4 on top of it. However, the FTGS training code used for pretraining (by the organizers) differs slightly from our own FTGS implementation. To avoid conflating these implementation discrepancies with the impact of our method, we instead report additional results on the Cinema scene from the MPEG dataset.
>
> Regarding long-sequence scenarios, to the best of our knowledge, there is currently no dataset with substantially more than 600 frames. As discussed above for complex-motion scenes, OMG4 can benefit from any advances in long-sequence reconstruction without redesigning the underlying architecture. We kindly ask your understanding that, given the current benchmark landscape, it is practically difficult to include experiments on longer sequences.

---

> > ### Author Response · Authors · 2025-11-30
> >
> > | Dataset / Method                                  | PSNR ↑ | SSIM ↑ | LPIPS ↓ | # Gaussians ↓ | Storage (MB) ↓ |
> > |---------------------------------------------------|--------|--------|---------|----------------|----------------|
> > | **Bartender (300 frames, MPEG; complex motion)**  | | | | | |
> > | RT4DGS  | 31.52  | 0.875  | 0.114   | 3,302,206 | 2028 |
> > | RT4DGS + OMG4 (L)  | 31.24  | 0.871 | 0.119 | 343,593 | 6.90 |
> > | RT4DGS + OMG4 (S) | 30.25 | 0.858 | 0.143 | 108,508 | 2.39  |
> > | FTGS | 32.14  | 0.895  | 0.074   | 2,000,000 | 488 |
> > |**FTGS + OMG4 (L)** |**32.08**|**0.890**|**0.089**|**1,235,010**|**23.38**|
> > |**FTGS + OMG4 (S)**|**31.96**|**0.886**|**0.091**|**710,519**|**13.71**|
> > |**GIFStream**|**31.28**|**0.883**|**0.095**|**433,415**|**19.50**|
> > |**Cinema (200 frames, MPEG; complex motion)**  | | | | | |
> > |**RT4DGS**|**27.87**|**0.837**|**0.164**|**3,653,693**|**2244**|
> > |**RT4DGS + OMG4 (L)**|**27.76**|**0.834**|**0.171**|**385,342**|**7.79**|
> > |**FTGS**|**33.25**|**0.930**|**0.062**|**2,000,000**|**488**|
> > |**FTGS + OMG4 (L)**|**33.15**|**0.925**|**0.072**|**1,057,245**|**20.42**|
> > |**FTGS + OMG4 (S)**|**32.90**|**0.921**|**0.079**|**630,535**|**12.35**|
> > |**GIFStream**|**27.52**|**0.847**|**0.128**|**481,788**|**26.93**|
> > | **Flame Salmon (N3DV; long sequences)** | | | | | |
> > | RT4DGS|28.52  | 0.910  | 0.092| 3,332,220 | 2047|
> > | RT4DGS + OMG4 (L)| 28.12  | 0.906| 0.179| 305,636| 6.16|
> > | RT4DGS + OMG4 (S)| 27.75  | 0.902| 0.217| 143,200| 3.06|
> > | FTGS| 30.22  | 0.938  | 0.057   | 1,000,000 | 244 |
> > |**FTGS + OMG4 (L)**|**29.95**|**0.936**|**0.067**|**359,716**|**7.18**|
> > |**GIFStream**|**28.04**|**0.909**|**0.077**|**1,306,910**|**55**|

---

### Official Review · Reviewer_ZNMV · 2025-10-29

**Soundness:** 2
**Presentation:** 3
**Contribution:** 3
**Rating:** 6
**Confidence:** 2

**Summary:**

This paper introduces OMG4 (Optimized Minimal 4D Gaussian Splatting), a novel framework for compact and high-fidelity dynamic scene representation. While prior 4D Gaussian Splatting (4DGS) approaches have achieved impressive real-time rendering results, they typically require millions of Gaussians, leading to significant memory and storage overheads.

**Strengths:**

- The proposed Static–Dynamic Score (SD-Score) is an elegant and effective contribution that unifies spatial and temporal importance estimation for Gaussian primitives.
- The multi-stage pipeline (Sampling → Pruning → Merging) is conceptually simple yet powerful, providing interpretability and modular extensibility.
- Extending Sub-Vector Quantization (SVQ) to the 4D domain and introducing a staged quantization strategy for stability is a non-trivial and meaningful extension of prior work.
- The paper is clearly written, with strong motivation and logical flow across sections.

**Weaknesses:**

(1) Experimental depth and comparison limitations:
- While the reported compression ratio and reconstruction quality are impressive, the paper could benefit from a more diverse set of baselines, including recent hybrid deformation-based approaches (e.g., ADC-GS or D-NeRF variants).
- It remains unclear how OMG4 scales with scene complexity or duration—e.g., whether compression quality degrades for highly non-rigid motions or long temporal spans.

(2) Computational cost and runtime:
- The paper focuses primarily on memory efficiency, but does not clearly state the training or optimization time overhead introduced by multi-stage pruning and merging.
- It would strengthen the work to clarify whether OMG4 maintains real-time rendering throughput post-compression, and how merging affects rendering speed and differentiability.

(3) Generality of SVQ extension:
- The adaptation of SVQ to 4D is interesting, but the explanation of why staged quantization improves stability is somewhat qualitative.
- A quantitative analysis (e.g., convergence behavior, quantization error curves) would substantiate this claim.

**Questions:**

See Weaknesses

---

> ### Author Response · Authors · 2025-11-21
>
> ## **Applying OMG4 to other 4D reconstruction models**
> Please refer to "Generality of the proposed method" part of General Response section. We initially applied our OMG4 to two baseline models, RT4DGS and FTGS, and additionally evaluated it on STG, another representative 4D reconstruction model, to further demonstrate the generality of our approach.
>
> ## **Applying OMG4 to scenes with complex motions or long temporal spans**
> Please refer to "Experiments on additional dataset" part of General Response section. We evaluated our method on four extra datasets, with non-rigid motions and longer temporal spans. Also, the qualitative results are provided at [link]( https://anonymous.4open.science/r/OMG4_Qualitative).
>
> ## **Computational cost and real-time rendering**
> We provide the computational cost analysis in "Training time analysis" part of the General Response section. Regarding real-time rendering, OMG4 can still enjoy real-time rendering. As shown in Tabs. 1–3 of the main paper, our method even achieves nearly 4.3x higher FPS than RT4DGS because it significantly reduces the number of Gaussians (57 vs. 246). When combined with FTGS, OMG4 leads to a modest FPS drop (from 129 to 91), but this still enables real-time rendering.
>
> Also, the “Gaussian Merging” module is fully differentiable, so the entire pipeline can be trained end-to-end. During training, we perform temporary merging for 2,000 iterations to learn the merging weights, which introduces only about 0.01 secondsof  additional time per iteration compared to training without merging. Once the weights are learned, we permanently merge Gaussians via a weighted sum, and the training-time overhead disappears. At test time, the Gaussians are already merged, so the merging operation does not incur any extra cost and does not affect FPS.
>
> ## **Staged SVQ**
> We conducted an ablation on the N3DV dataset comparing our staged SVQ strategy (3D SVQ followed by 4D SVQ) against applying 3D and 4D SVQ simultaneously (“1-stage SVQ”). The results are summarized in the table below. Our staged SVQ yields consistently better reconstruction quality across different metrics, while maintaining a comparable storage footprint. This confirms that the proposed staged design is more effective at exploiting the complementary roles of 3D and 4D quantization than the simultaneous variant.
> | Method               | PSNR ↑ | SSIM ↑ | LPIPS ↓ | # Gaussians ↓ | Storage (MB) ↓ |
> |----------------------|--------|--------|---------|----------------|----------------|
> | 1-stage SVQ          | 31.57  | 0.940  | 0.061   | 171,380        | 3.64           |
> | 2-stages SVQ (Ours)  | 31.80  | 0.941  | 0.059   | 171,136        | 3.61           |

---

### Author Response · Authors · 2025-11-21

We are grateful to all the reviewers for their reviews and invaluable feedback. Before responding to each review in detail, we first address several common questions in this general response. For better assessment, we provide additional qualitative results at [link]( https://anonymous.4open.science/r/OMG4_Qualitative).
## **Generality of the proposed method**
As shown in Tab. 3 in the manuscript, we also applied the proposed method to FreeTimeGS (FTGS) [Ref 1]. We reduced the storage requirement of FTGS by 90\%, demonstrating the generality of OMG4. To further secure the robustness of OMG4, we applied our method to SpaceTimeGaussian (STG) [Ref 2] and evaluated it on the N3DV dataset. The aggregated results are reported in the table below. On STG, OMG4 reduces the model size by 88\% with comparable reconstruction quality. As OMG4 aims to effectively reduce the number of primitives required for modeling the scene, it can be broadly adopted to various approaches, including RealTime4DGS, FTGS, and STG.
| Method       | PSNR ↑ | SSIM ↑ | LPIPS ↓ | # Gaussians ↓ | Storage (MB) ↓ |
|-------------|--------|--------|---------|----------------|----------------|
| STG         | 32.00      | 0.948      | 0.046       | 1,289,577              | 200              |
| STG + OMG4  | 31.72      | 0.945      | 0.049       | 863,716              | 24.38              |

[Ref 1] Wang, Yifan, et al. "FreeTimeGS: Free Gaussians at Anytime and Anywhere for Dynamic Scene Reconstruction." Proceedings of the IEEE/CVF Conference on Computer Vision and Pattern Recognition. 2025.

[Ref 2] Li, Zhan, et al. "Spacetime gaussian feature splatting for real-time dynamic view synthesis." Proceedings of the IEEE/CVF Conference on Computer Vision and Pattern Recognition. 2024.

---

> ### Author Response · Authors · 2025-11-21
>
> ## **Experiments on additional dataset**
> As shown in Tab. 2 in the main paper, we initially conducted experiments on *Bartender* scene of MPEG dataset, which exhibits more complex and non-rigid motions than N3DV dataset. However, in line with prior work, we used only the first 65 frames. In this discussion phase, we therefore conduct additional experiments using the full sequence (300 frames) to better demonstrate the effectiveness of the proposed method on complex, long-term motion. Moreover, as suggested, we further evaluate our approach on additional datasets with more complex motions and longer sequences that were not included in the main paper, including MeetRoom [Ref 1], SIGGRAPH ASIA 1st Volumetric Video Challenge (SA-VVC) [Ref 2], and *Flame Salmon* scene of N3DV data with 600 frames--twice the length of the sequence used in the main paper. The overall statistics of these datasets are summarized in the table below.
> | Dataset               | # Scenes | # Frames | Resolution   |
> |-----------------------|----------|----------|--------------|
> | MeetRoom              | 3        | 300      | 1280 x 720   |
> | SA-VVC                | 1        | 300      | 4019 x 2200  |
> | Bartender (MPEG)      | 1        | 300      | 1920 x 1080  |
> | Flame Salmon (N3DV)   | 1        | 600      | 1352 x 1014  |
>
> We reported the results of applying OMG4 on new dataset in the following table. We also provide links (at the top) where reviewers can directly verify their visual complexity and challenging motion patterns. As SA-VVC dataset contains particularly challenging motions (e.g., dancing), RT4DGS could not successfully reconstruct the scene, thus we only reported the results of FTGS. Despite the difficulty of the dataset, our method can produce stable and high-quality reconstructions, indicating that it can robustly handle complex, highly non-rigid motion and long temporal span.
>
> | Dataset / Method                                  | PSNR ↑ | SSIM ↑ | LPIPS ↓ | # Gaussians ↓ | Storage (MB) ↓ |
> |---------------------------------------------------|--------|--------|---------|----------------|----------------|
> | **MeetRoom (complex motion)**                     |        |        |         |                |                |
> | RT4DGS                                            | 26.60  | 0.910  | 0.075   | 3,102,842      | 1906           |
> | RT4DGS + OMG4 (L)                                 | 26.81  | 0.905  | 0.071   | 360,533        | 7.19           |
> | RT4DGS + OMG4 (S)                                 | 26.65  | 0.904  | 0.075   | 163,893        | 3.45           |
> | FTGS                                              | 28.25  | 0.922  | 0.064   | 2,000,000      | 488            |
> | FTGS + OMG4 (L)                                   | 28.30  | 0.924  | 0.071   | 539,105        | 10.65          |
> | FTGS + OMG4 (S)                                   | 28.26  | 0.924  | 0.075   | 277,209        | 5.60           |
> | **Bartender (300 frames, MPEG; complex motion)**  |        |        |         |                |                |
> | RT4DGS                                            | 31.52  | 0.875  | 0.114   | 3,302,206      | 2028           |
> | RT4DGS + OMG4 (L)                                 | 31.24  | 0.871  | 0.119   | 343,593        | 6.90           |
> | RT4DGS + OMG4 (S)                                 | 30.25  | 0.858  | 0.143   | 108,508        | 2.39           |
> | FTGS                                              | 32.14  | 0.895  | 0.074   | 2,000,000      | 488            |
> | FTGS + OMG4 (L)                                   | 30.51  | 0.853  | 0.152   | 592,571        | 11.41          |
> | FTGS + OMG4 (S)                                   | 30.12  | 0.841  | 0.168   | 330,779        | 6.50           |
> | **SIGGRAPH VVC dataset (complex motion)**  | | | | | |
> | FTGS                                              | 30.34  | 0.956  | 0.076   | 4,757,948      | 581            |
> | FTGS + OMG4 (L)                                   | 28.31  | 0.948  | 0.099  | 1,512,833 | 32.45 |
> | **Flame Salmon (N3DV; long sequences)** | |  | | | |
> | RT4DGS                                            | 28.52  | 0.910  | 0.092   | 3,332,220      | 2047           |
> | RT4DGS + OMG4 (L)                                 | 28.12  | 0.906  | 0.179   | 305,636   | 6.16           |
> | RT4DGS + OMG4 (S)                                 | 27.75  | 0.902  | 0.217   | 143,200        | 3.06           |
> | FTGS                                              | 30.22  | 0.938  | 0.057   | 1,000,000 | 244 |
> | FTGS + OMG4 (L)                                   | 29.28  | 0.920  | 0.103   | 621,685 | 12.20 |
> | FTGS + OMG4 (S)                                   | 28.76  | 0.909  | 0.129   | 157,746 | 3.29 |
>
> [Ref 1] Li, Lingzhi, et al. "Streaming radiance fields for 3d video synthesis." Advances in Neural Information Processing Systems 35 (2022): 13485-13498.
>
> [Ref 2] https://www.4dv.ai/research/sig-asia2025-volumetric-video-challenges#challenges

---

> ### Author Response · Authors · 2025-11-21
>
> ## **Training time analysis**
> Below table reports the optimization time of our method on N3DV dataset, measured with a RTX 4090 GPU. We train OMG4 for a total of 11,000 iterations, and, before optimization, we perform a preprocessing step that computes view-space and temporal gradients. The row “OMG4” therefore includes both this preprocessing and the subsequent optimization while “OMG4*” excludes the gradient computation and reports only the pure optimization time. Thanks to our effective reduction of the Gaussian count, the per-iteration optimization time decreases from 0.31 s/iter for the original pretraining stage to 0.157 s/iter for OMG4*, i.e., nearly a 2× speedup. The lower part of the table further decomposes the wall-clock time of each module. “Gradient Computing” corresponds to the one-time cost of preprocessing.
>
> “Gaussian Sampling” and “Gaussian Pruning” are both run for 1,000 iterations. “Gaussian Pruning” is applied after “Gaussian Sampling” has already reduced the number of points, which explains its slightly lower per-iteration time. For “Gaussian Merging”, we separately list the time for cluster computation (“Calc. Clusters”) and the subsequent optimization. During merging, Gaussians are temporarily combined for rendering, which introduces a small overhead of about 0.01 s per iteration; in the second merging stage, the number of Gaussians is already reduced, so both clustering and optimization are faster than in the first stage.
>
> In the “Attribute Compression” block, we show that using a small MLP for attribute encoding has a negligible impact on runtime and that the additional stages with SVQ-3D and SVQ-4D introduce only a modest increase in per-iteration time. After optimization is done, decoding the compressed model takes approximately 8 seconds, which is a one-time cost. Overall, these results demonstrate that our pipeline substantially reduces optimization time per iteration while keeping each compression and merging component computationally lightweight.
>
> | Method               | Sub-module              | # Iterations | Time (min) | Time / iter (sec) |
> |----------------------|-------------------------|--------------|-----------:|-------------------:|
> | Pretraining          | -                       | 30,000       |   155.05   | 0.310             |
> | OMG4                 | -                       | 11,000       |    45.17   | 0.246             |
> | OMG4*                | -                       | 11,000       |    28.83   | 0.157             |
> |                      |                         |              |            |                   |
> | Gradient Computing   | -                       | -            |    16.35   | -                 |
> | Gaussian Sampling    | -                       | 1,000        |     2.47   | 0.148             |
> | Gaussian Pruning     | -                       | 1,000        |     2.41   | 0.144             |
> | Gaussian Merging 1   | Calc. Clusters          | -            |     0.83   | -                 |
> | Gaussian Merging 1   | Optimization            | 1,000        |     2.57   | 0.154             |
> | Gaussian Merging 2   | Calc. Clusters          | -            |     0.59   | -                 |
> | Gaussian Merging 2   | Optimization            | 1,000        |     2.44   | 0.146             |
> | Attribute Compression| MLP                     | 5,000        |    12.14   | 0.146             |
> | Attribute Compression| MLP + SVQ 3D            | 1,000        |     2.57   | 0.154             |
> | Attribute Compression| MLP + SVQ 3D + SVQ 4D   | 1,000        |     2.83   | 0.170             |

---

### Author Response · Authors · 2025-12-03

### Summary of Reviewer Ratings

We thank all reviewers for their feedback. A brief overview of the rebuttal is as follows:

| **Reviewer** | **Initial Score** | **Post-Rebuttal Score** | **Note** |
|--------------|------------------|-------------------------|----------|
| **Reviewer ZNMV** | 6 | 6 (Waiting for response) | Provided results on the requested experiments: 1) Applying OMG4 to other 4D baselines 2) and scenes with complex motions or long sequences  3) Computational cost analysis 4) Ablation on staged SVQ |
| **Reviewer jaTu** | 6 | 6 (Waiting for response) | Provided results on challenging datasets which **the reviewer noted would raise the score to 8** if demonstrated: 1) Dataset with complex motion and long sequences 2) Comparison with SOTA model
| **Reviewer 3wjx** | 4 | **8 (Raised)** | Explicitly raised score; concerns fully addressed: 1) Robustness to hyperparameters. 2) Zero-shot dynamic–static distinguishment |

---

### Meta-Review · Area_Chair_sDeB · 2026-01-13

**Summary:**

The initial ratings of this paper are: 6 (ZNMV, Confidence: 2), 6 (jaTu, Confidence: 5) , 4 (3wjx, Confidence: 4). However, I found the rebuttal responses not very satisfactory. ZNMV with low confidence did not respond. Nevertheless, his comments are valid. The reviewer jaTu was not fully convinced by the initial responses and results. The authors then updated their results, indicating that the training was not completed. However, the authors also changed one test sequence, which leaves a sense of cherry-picking. The reviewer 3wjx would increase his rating. My recommendation is to wait for more stable and comprehensive results.

Reviewer ZNMV (6: marginally above the acceptance threshold. But would not mind if paper is rejected; 2: You are willing to defend your assessment)

o	While the reported compression ratio and reconstruction quality are impressive, the paper could benefit from a more diverse set of baselines, including recent hybrid deformation-based approaches (e.g., ADC-GS or D-NeRF variants).

o	It remains unclear how OMG4 scales with scene complexity or duration—e.g., whether compression quality degrades for highly non-rigid motions or long temporal spans.

[AC: Additional results were generated during the rebuttal period for long sequences (e.g. MeetRoom, SA-VVC, Bartender, Flame Salmon) that were not initially included in the paper. However, a considerable PSNR drop (1-2 dB) is observed across several test cases when OMG4 is applied, although the storage size with OMG4 is much reduced. It then validates the doubt from the reviewer. This view is also shared by another reviewer jaTu during the rebuttal period. Note that the authors later updated their results, showing much improved performance. Their explanation: “Initially, we conducted all FTGS + OMG4 experiments with 5,000 iterations of MLP optimization in the Attribute Compression stage. However, we later found that the MLP had not fully converged for FTGS if the dataset shows complex motion or contains long sequences. Therefore, only for the FTGS + OMG4 case, we increased the number of MLP training iterations from 5,000 to 12,000.” However, this then leads to the training time concern. In addition, the authors later chose a different sequence to replace SIGGRAPH VVC. My recommendation is to wait for more stable and comprehensive results.]

o	The paper focuses primarily on memory efficiency, but does not clearly state the training or optimization time overhead introduced by multi-stage pruning and merging.

[AC: Additional results were generated during the rebuttal period to provide a breakdown analysis of training time. The proposed method OMG4 reduces the time per iteration from 0.31s to 0.24s. In this 0.24s, 0.16s is dedicated to optimization (sampling, pruning and merging). I however feel that the results are indirect in terms of assessing the training overhead. It would be instructive to compare the training time needed for the proposed method to achieve a similar image quality level to the baseline method (e.g. RealTime-4DGS and OMG).]

o	It would strengthen the work to clarify whether OMG4 maintains real-time rendering throughput post-compression, and how merging affects rendering speed and differentiability.

[AC: The proposed method does not have any impact on real-time rendering. Instead, the rendering FPS may be improved because of having much less Gaussians.]

o	The adaptation of SVQ to 4D is interesting, but the explanation of why staged quantization improves stability is somewhat qualitative. A quantitative analysis (e.g., convergence behavior, quantization error curves) would substantiate this claim.

[AC: Additional quantitative results are provided. No issue here.]

Reviewer jaTu (6: marginally above the acceptance threshold. But would not mind if paper is rejected; 5: You are absolutely certain about your assessment.)

o	The experiments are too limited — most evaluations are conducted only on the N3DV dataset. Testing on large-scale dynamic scenes such as NVIDIA[1], Dynamic3DGS[2], or VRU [3] would make the results much more convincing.

[AC: The same concern is raised by ZNMV. This reviewer is referred to the same set of experimental results.]

o	The proposed compression and pruning strategies are all based on 4DGS; however, 4DGS has inherent limitations, such as its inability to handle fast motion or long-sequence dynamic videos. Therefore, this method is somewhat restricted in its applicability.

[AC: The authors indicate that OMG4 achieves strong compression with only minor visual degradation across three baselines, RT4DGS, FTGS, and STG. I would NOT consider a PSNR drop of 1-2dB in some test cases (FTGS vs. FTGS+OMG4 (L/S) on Flame Salmon, Bartender and SIGGRAPH VVC) to be minor visual degradation, although the compression ratio is high. Their explanation: “Initially, we conducted all FTGS + OMG4 experiments with 5,000 iterations of MLP optimization in the Attribute Compression stage. However, we later found that the MLP had not fully converged for FTGS if the dataset shows complex motion or contains long sequences. Therefore, only for the FTGS + OMG4 case, we increased the number of MLP training iterations from 5,000 to 12,000.” However, this then leads to the training time concern. In addition, the authors later chose a different sequence to replace SIGGRAPH VVC. My recommendation is to wait for more stable and comprehensive results.]

o	How long does the model take to train?

[AC: The same concern is raised by ZNMV. This reviewer is referred to the same set of experimental results. As said, the results provided are indirect. It would be instructive to compare the training time needed for the proposed method to achieve a similar image quality level to the baseline method (e.g. RealTime-4DGS and OMG).]

[AC: The authors argue that “to the best of our knowledge, there is currently no dataset with substantially more than 600 frames. As discussed above for complex-motion scenes, OMG4 can benefit from any advances in long-sequence reconstruction without redesigning the underlying architecture. We kindly ask your understanding that, given the current benchmark landscape, it is practically difficult to include experiments on longer sequences.” This is NOT true actually. The ENeRF dataset (https://zju3dv.github.io/enerf/) has 4 sequences of 1200 frames.]

o	Is the model initialized using Gaussian points from all frames to represent the entire scene, followed by the sampling, merging, and pruning strategies? If so, wouldn’t this result in high GPU consumption and require long training times — possibly several hours?

[AC: This comment is not addressed explicitly.]

[AC: Overall, this reviewer indicates that he is NOT convinced by the authors’ results.]

Reviewer 3wjx [4: marginally below the acceptance threshold. But would not mind if paper is accepted; 4: You are confident in your assessment, but not absolutely certain.]

o	Although the proposed method provides superior compactness, the proposed components in the method are not very impressive in the aspect of novelty. The Gaussian importance score, MLP based color coding, and post-processing compressors are upgraded from 3DGS compression techniques or previously established in other 4D Gaussian compression works. These make the novelty of this work weaker, and the authors fail to distinguish their designs from the previous techniques. My own opinion is that the author can emphasize more on the generality of the proposed method: The authors can establish the whole pipeline as a more robust and general post-processing progress to all (or most of) the previous 4DGS baselines, and investigate how to specifically implement the proposed method to each distinct baselines and fit the special property of each method.

[AC: The same concern is raised by ZNMV. This reviewer is referred to the same set of experimental results. The authors do not seem to have addressed the novelty aspect.]

o	Another concern is a highly engineered pipeline with many hyperparameters. The whole pipeline includes multiple thresholds, quantiles, grid sizes and merge / iteration budgets. These elements raise concerns on the robustness of the proposed method. Naturally, some scenes require high quantity of Gaussians for representation, while some other may not. Some scenes contain complex lattice or textures. The discussion or ablation on these issues should be provided for a solid presentation.

[AC: The authors commented that they did not fine-tune the hyperparameters for each scene, but kept the same hyperparameters across all scenes in each dataset.]

o	Some previous 4D reconstruction methods use zero-shot dynamic-static distinguishment [R1, R2]. Can they benefit the proposed pipeline? These methods should also be considered or discussed.

[AC: Additional results are provided.]

[AC: Overall, the reviewer was satisfied with the responses and expressed that the score will be increased.]

**Reviewer Concerns:**

See my comments in the summary section.

**Reviewer Scores:**

The initial ratings of this paper are: 6 (ZNMV, Confidence: 2), 6 (jaTu, Confidence: 5) , 4 (3wjx, Confidence: 4). However, I found the rebuttal responses not very satisfactory. ZNMV with low confidence did not respond. Nevertheless, his comments are valid. The reviewer jaTu was not fully convinced by the initial responses and results. The authors then updated their results, indicating that the training was not completed. However, the authors also changed one test sequence, which leaves a sense of cherry-picking. The reviewer 3wjx would increase his rating. My recommendation is to wait for more stable and comprehensive results.

---

### Decision · Program_Chairs · 2026-01-26

Reject